



# The transition to new ozone absorption cross-sections for Dobson and Brewer total ozone measurements

Karl Voglmeier[1], Voltaire A. Velazco[1], Luca Egli[2], Julian Gröbner[2], Alberto Redondas[3], Wolfgang Steinbrecht[1]

[1]Deutscher Wetterdienst, Meteorological Observatory Hohenpeißenberg, 82383, Germany
[2]Physikalisch-Meteorologisches Observatorium Davos (PMOD/WRC), Davos Dorf, 7260, Switzerland
[3]Izaña Atmospheric Research Center, Agencia Estatal de Meteorología, Tenerife, 38001, Spain

*Correspondence to*: Karl Voglmeier (karl.voglmeier@dwd.de)

**Abstract.** Comparison of total ozone column (TOC) measurements from ground-based Dobson and Brewer spectrophotometers and from various satellite instruments generally reveals seasonally varying differences of a few percent. A large part of these differences has been attributed to the operationally used Bass & Paur ozone cross-sections and the lack of accounting for varying stratospheric temperatures in the standard total ozone retrieval for Dobson. This paper demonstrates how the use of new ozone absorption cross sections from the University of Bremen (Weber et al., 2016), as recommended by the committee on Absorption Cross-Sections of Ozone, the application of appropriate slit functions, especially for the Dobson instrument (Bernhard et al. 2005), and the use of climatological values for the effective ozone layer temperature ($T_{eff}$), e.g. from TEMIS, essentially eliminate these seasonally varying differences between Dobson and Brewer total ozone data. Applying this approach to the existing global network of Dobson spectrometers will reduce the uncertainty of their total ozone data, from previously 3 to 4% to better than 2.0% at most locations.

## 1   Introduction

Ground based total ozone column (TOC) measurements can be obtained by a large number of methods, but within the framework of the Global Atmosphere Watch program (GAW) of the World Meteorological Organization (WMO), Dobson and Brewer spectrophotometer measurements are considered as reference observations. Worldwide, a large number of Dobson and Brewer instruments are used, and TOC measurements are routinely reported to the World Ozone and Ultraviolet Radiation Data Centre (WOUDC). Dobson spectrometers were developed in the 1920s and have been used for continuous measurements for decades, e.g. since 1926 in Switzerland (Stübi et al., 2021). Brewer spectrometers have been widely used since the 1980s. Both instruments have a good long-term stability and precision (Stübi et al., 2017, 2021), and many research groups at different locations perform TOC measurements with Brewer and Dobson side by side. A seasonally varying  systematic difference (or bias) between the two instruments has long been recognized (Kerr et al., 1988; Scarnato et al., 2010; Vanicek, 2006; Vaníček





et al., 2012). Seasonally varying differences (biases) have also been found in the comparison of Dobson and Brewer total ozone with data from satellite instruments (Koukouli et al., 2015, 2016).

Much of these biases has been attributed to the operationally used ozone absorption cross sections at fixed effective ozone temperature (Bass and Paur, 1985; Komhyr and Evans, 2008), which neglects the temperature sensitivity of these absorption cross sections (Koukouli et al., 2016; Köhler et al., 2018).

From 2008 to 2015, the "Absorption Cross-Section of Ozone" (ACSO) committee evaluated a number of newly measured ozone absorption cross-section data sets and recommended to use the data of Serdyuchenko et al. (2014) and Gorshelev et al. (2014) (further on denoted as SG14) for ground-based TOC measurements (Orphal et al., 2016). However, in the operational community, those recommendations are still not applied routinely.

Recently, Gröbner et al. (2021) and Redondas et al. (2014) have retested different sets of ozone absorption cross sections and
also accounted for their temperature dependence using ozone effective temperatures ($T_{eff}$) from modelled or measured data. In both studies, using effective ozone temperature and the SG14 absorption cross section set reduced the difference between Brewer and Dobson total ozone data significantly. For the comparison with various satellite instruments, Koukouli et al. (2016) also showed substantial improvements when applying a linear Teff-dependent correction to the available Dobson data.

The purpose of this study is to check and further update the findings of Redondas et al. (2014), Orphal et al. (2015), and
Gröbner et al. (2021). In addition, we address the following important points:

- We test the additional Weber et al. (2016) ozone absorption cross-section dataset, which is similar to Serdyuchenko et al. (2014), but has better quantification of uncertainty and improved polynomial fitting coefficients for temperature dependence.

- We test two new ozone absorption cross-section datasets (Gorshelev et al., 2017; linked to Serdyuchenko et al. 2014,
but with updated coefficients for temperature dependence) and Birk and Wagner (2021).

- We test different ways to account for the Dobson slit functions, which describe the instrument response to radiation at wavelengths near the nominal central wavelengths.

- We check ways to obtain ozone effective temperature and investigate their impact on TOC retrieval, including the comparison of daily effective temperature values with climatological values.

- We examine the effect of applying new temperature-dependent absorption cross-section datasets at different locations of Dobson and Brewer instruments worldwide.

- We provide recommendations how to easily implement the new temperature-dependent ozone absorption coefficients in the operational Dobson TOC network.

The ultimate goal of this study is to pave the way for implementing the new temperature-dependent absorption cross sections
in historical and in operational retrievals for ground-based total ozone column (TOC) measurements.





## 2 Total ozone column measurements and retrieval

### 2.1 Measurement principle

Atmospheric concentration measurements by both instrument types are based on Beer-Lambert's law:

$$I(\lambda) = I_0(\lambda)exp^{-\tau(\lambda)\mu}$$ (Eq. 1)

where $I_0$ and $I$ are the wavelength dependent solar irradiance at the top of the atmosphere and at the surface, respectively, $\tau$ is the optical depth of the atmosphere, and $\mu$ is the relative air mass (slant path through the atmosphere).

In the wavelength region between 300 and 345 nm, where both instruments measure TOC, ozone molecules are the main absorber of solar irradiation. $SO_2$ absorption in this wavelength region can occur, but is typically small at most locations, and can only be quantified by the Brewer instrument. Thus, the results shown in this study are limited to unpolluted air, where $SO_2$

values from Brewer instruments are low (< 1.0 DU).

Taking only into account the absorption of solar irradiance by the ozone molecules, and correcting for Rayleigh scattering effects, Eq. (1) can be expressed as:

$$I(\lambda) = I_0(\lambda)exp^{-(TOC\alpha(\lambda)\mu_{O_3}+\beta(\lambda)\frac{p_S}{p_0}m_R)}$$ (Eq. 2)

where TOC is the vertical total column of ozone, $\mu_{O_3}$ is the relative air mass for ozone, $\alpha$ represents the wavelength dependent ozone absorption coefficient, $\beta$ is the wavelength dependent Rayleigh extinction coefficient, $p_S$ and $p_0$ are the atmospheric pressure at the station and at sea level, respectively, and $m_R$ is the Rayleigh air mass.

Rearranging Eq 2. gives

$$TOC\ \alpha(\lambda)\mu_{O_3} = ln(I_0(\lambda)) - ln(I(\lambda)) - \beta(\lambda)\frac{p_S}{p_0}m_R$$ (Eq. 3)

This equation is valid for any wavelength. If measurements are taken at e.g. two wavelengths $\lambda_1$ and $\lambda_2$, the resulting two Eq. (3) can be subtracted which gives

$$TOC\ (\alpha(\lambda_1) - \alpha(\lambda_2))\mu_{O_3} = ln(I_0(\lambda_1)) - ln(I_0(\lambda_2)) - (ln(I(\lambda_1)) - ln(I(\lambda_2)) - (\beta(\lambda_1) - \beta(\lambda_2))\frac{p_S}{p_0}m_R$$ (Eq. 4)

This approach can be expanded to more wavelengths, and to any linear combination of the resulting Eq. (3).

Consequently, TOC can be calculated from linear combinations of measurements at different wavelengths ($\lambda_i$):

$$TOC = \frac{\Delta F_0 - \Delta F - \Delta\beta\frac{p_S}{p_0}m_R}{\Delta\alpha\mu_{O_3}}$$ (Eq. 5)



Where $\Delta F_0$ and $\Delta F$ are linear combinations of $\ln(I_0(\lambda_i))$ and $\ln(I(\lambda_i))$, and $\Delta\alpha$ and $\Delta\beta$ are the corresponding linear combinations of ozone absorption cross sections $\alpha(\lambda_i)$ and Rayleigh extinction cross sections $\beta(\lambda_i)$. It's worth noting that in our study, we applied the nominal Rayleigh scattering coefficients for both instruments from the standard algorithm (Bates, 1984; Komhyr and Evans, 2008).

$$\Delta F_X = \sum_{i=1}^{n} w_i \ln(I_X(\lambda_i)) \tag{Eq. 6}$$

$$\Delta\beta = \sum_{i=1}^{n} w_i \beta_i \tag{Eq. 7}$$

$$\Delta\alpha = \sum_{i=1}^{n} w_i \alpha_i \tag{Eq. 8}$$

Potential aerosol influences are minimized by using multiple wavelengths, e.g. four single slit measurements with appropriate weights in the case of the Brewer instrument (Redondas et al., 2014), or two-wavelength pairs (typically AD, or CD) in the case of the Dobson instrument (Komhyr and Evans, 2008).The weighting coefficients $w_i$ for the sums in Equations 6-8, for both Dobson and Brewer, are given in the last columns of Tables 1 and Table 2.

## 2.2    Brewer and Dobson spectrophotometers measurements at Hohenpeissenberg

The Brewer spectrometer is fully automated. In ozone mode it measures solar irradiance at six nominal wavelengths in the UV range, from 303.2 to 320.1 nm, quasi-simultaneously. This is achieved by using a slit mask in combination with a holographic grating and a photomultiplier tube. The calculation of TOC following Eq. (5) uses measurements only at the four longest wavelengths of the six. A detailed description of the Brewer instrument can be found in Brewer ( 1973), Kerr et al. (1985), and Redondas et al. (2018).

Two Brewer instruments are currently in operation at the Meteorological Observatory Hohenpeissenberg (MOHp). Brewer010 is a single-monochromator Brewer MKII that has continuously measured TOC since 1983. The MKIII double-monochromator Brewer226 has been continuously measuring TOC since 2015. Both instruments are calibrated once a year by comparing them with the reference travelling standard single-monochromator Brewer017, operated by International Ozone Service (IOS).

The Dobson spectrometer measures TOC by comparing the relative intensities at two of three wavelength pairs in the UV wavelength range from 305.5 to 339.9 nm. These wavelength pairs are referred to as A, C, or D (the B pair is normally not used). Each pair compares solar irradiation in a "short" wavelength band that is highly absorbed by ozone, to solar irradiation in a "long" wavelength band that is less affected by ozone. For each measurement, an optical attenuator (a.k.a. "wedge"), is gradually adjusted to reduce the higher light intensity at the "long" wavelength, until it is equal to the lower light intensity at the "short" wavelength. With the information on the exact ratio of the long-to-short wavelength intensities, TOC values are then determined using the double-ratio of two pair measurements following Eq. (5). Typically, the A and D pairs are the most widely used pairs.





MOHp has been using Dobson104 operationally since 1968, with emphasis on direct sun AD measurements. For this study, we will exclusively use AD measurements. Typically, these measurements are performed from Monday to Friday only, resulting in approximately 1200 measurements per year. Dobson104 undergoes regular calibration by comparison with the Dobson reference instrument Dobson064, maintained by the Regional Dobson Calibration Centre Europe and also located at the MOHp. The most recent calibration of Dobson104 was in 2019.

Internal stray light can affect both Brewer and Dobson instruments, as noted by Karppinen et al. (2015), Moeini et al. (2019) and Scarnato et al. (2009). Typically, the impact of stray light manifests as lower TOC values at high ozone slant pass values. This means, at low sun elevation angles, and high TOC values, retrieved TOC by these instruments is underestimated as indicated by Bais et al. (1996) and Redondas et al. (2014). However, double monochromator Brewers are much better equipped to suppress stray light, resulting in minimal straylight effects.

As mentioned, both types of instruments are reference measurement systems for ground-based TOC measurements in the GAW program. Thus, they should yield similar TOC values when measuring side by side. For comparison between the two instrument types in this study, the following data processing filters were applied:

- Time period between Dobson and Brewer measurements $\leq 15$ min
- Multiple Dobson measurements within a time interval of $\leq 15$ min were averaged
- Ozone airmass $\leq 3.6$
- $SO_2$ from Brewer $\leq 1.0$ DU
- Time period May 2008 – December 2021 for the comparison between Dobson104 and Brewer010
- Time period June 2018 – December 2021 for the comparison between Dobson104 and Brewer226

In total, we used 8135 measurements to compare Dobson104 with Brewer 010, around 1300 taken during the winter season and 2300 taken during summer. For the comparison of Brewer226 with Dobson104, we used 2250 measurements, around 420 taken during winter and 760 taken during summer.

Figure 1 shows the typical distinct seasonal cycle in the difference of TOC values from Brewer and Dobson. Throughout the summer months, both instruments give very similar total ozone columns. During the winter months notable differences arise, and the Dobson typically reports 1 to 2% smaller TOC than Brewer010 (up to 3% for Brewer226, see Supplement). In the annual average, this results in a difference of about 1% between Brewer010 and Dobson104 TOC, and about 1.4% between Brewer226 and Dobson104. Very similar differences are reported for other locations and instruments (e.g. Gröbner et al., 2021; Redondas et al., 2014).

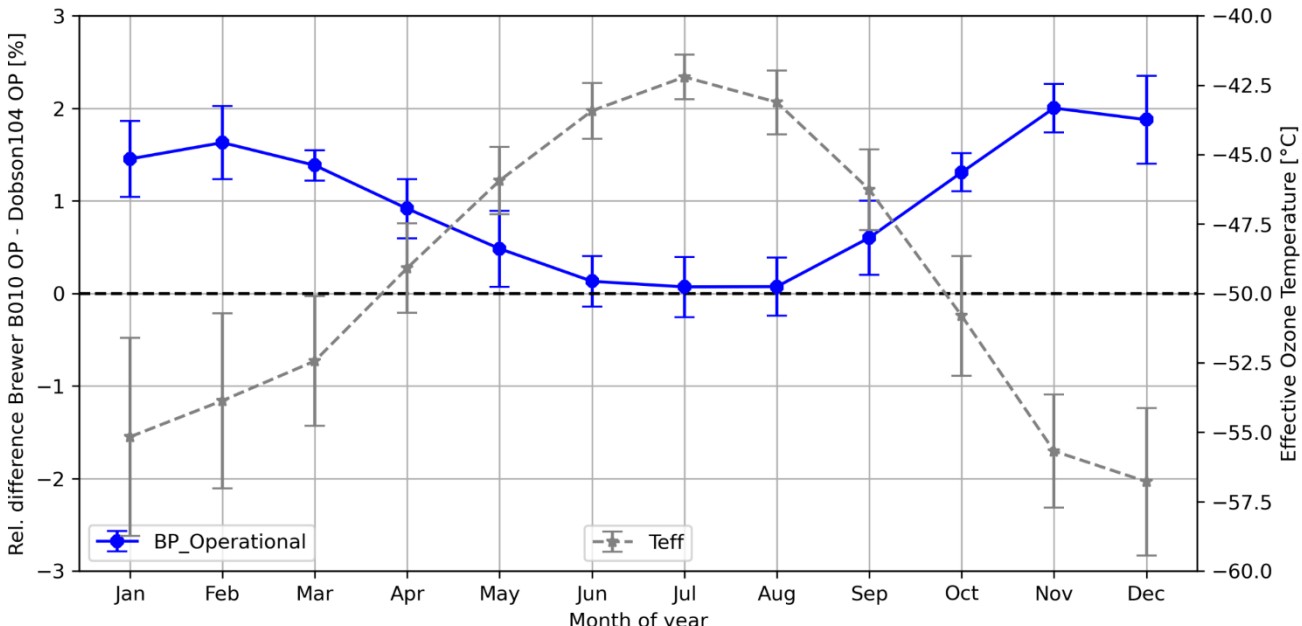

**Figure 1: Average monthly mean (1990 – 2020) differences between Brewer010 and Dobson104 (blue line), using the operationally used standard Bass and Paur ozone absorption cross sections. The dashed grey line gives the average monthly means of effective**
**ozone temperature ($T_{eff}$). The error bars represent the standard deviation (1σ, 1990 - 2020).**

## 2.3    Slit weighting functions

Since both Dobson and Brewer measure with limited spectral resolution, it is necessary to consider the spectral variation of the ozone cross-section over the wavelength bands covered by the instrument. Typically, the varying sensitivity in each wavelength band is called the slit-function. The high-resolution ozone cross-section needs to be averaged over these slit

functions, yielding effective ozone cross-section for each measured wavelength or wavelength pair.

### 2.3.1    Dobson

The Dobson network uses two slightly different parametrizations for the typical slit functions. Both parametrizations are based on the measured slits of Dobson083 (Komhyr et al., 1993), the world primary standard, which are quite similar to recently measured slit functions of Dobsons using tuneable lasers (Köhler et al., 2018). The Dobson Operations Handbook (Komhyr

and Evans, 2008) assumes a triangular slit function for the three short wavelength slits. Bernhard et al. (2005) assume trapezoids for the same short wavelength bands (Figure 2). The long wavelength slits are parametrized as trapezoids in both approximations. In addition to these standard parametrizations, slit functions have also been measured directly using a tuneable and portable radiation source (TuPS), developed in the joint research project EMRP ENV59 ATMOZ (Šmíd et al., 2021). The slits of the Dobson104 were measured with TuPS in October 2017, and the resulting slit functions were also tested here. They

are shown in Fig. 2 (red lines). Especially for the short wavelength slits, e.g. slit A1 in Fig. 2a, it is important to consider the





relatively wide wings of the TuPS slit function (dotted red line in Fig. 2a). Especially at the short wavelengths, below 304 nm in Fig. 2a, the large ozone cross sections bring a considerable contribution to the effective ozone cross section, which is integrated over the entire slit function (see also Gröbner et al., 2021).

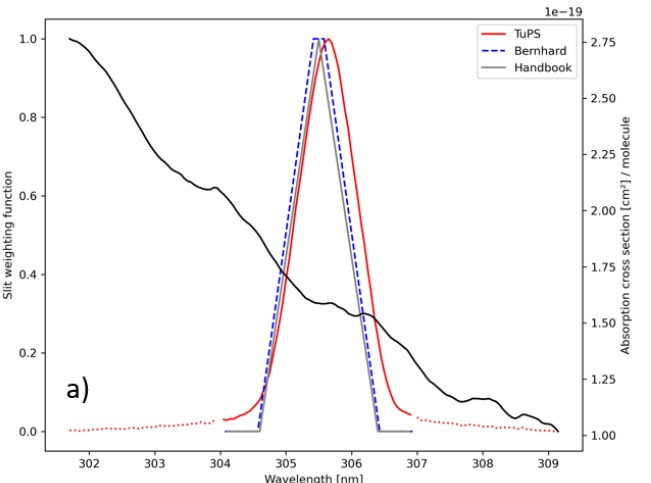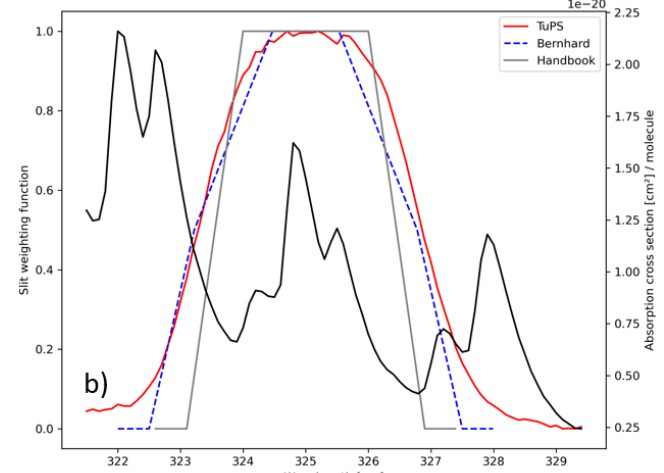


**Figure 2: Parametrized and measured slit weighting functions. Grey line: Dobson Operations Handbook. Blue line: Bernhard et al. (2005). Red line: measured slit functions (TuPS) for the Dobson104. TuPS measurements of Dobson104 were combined with TuPS measurements from Dobson101 (red dotted line, only for slits A1 and D1) to extend the spectral range of the slit function. Left panel (a) for the short wavelengths, and right panel (b) for the long wavelengths of the A wavelength pair. The black line gives the SG16**
**absorption cross section at a temperature of -55 °C.**

The slit parameters (central wavelength, FWHM of each slit, base and top for Bernhard slit approximation) for Dobson104 are shown in Table 1.

**Table 1: Central wavelength (mean, nm), full width at half maximum (FWHM, nm), and base (nm) and top (nm) for the Bernhard**
**slit approximation, for the individual slit functions of Dobson104. The nominal values were obtained from the Dobson Operations Handbook (Komhyr and Evans, 2008). The Bernhard values were obtained from table 1 in Bernhard et al. (2005). The weights $w_i$ (in the last column) are required to calculate the final absorption coefficients as described in section 2.6.**

| Slit | D104 Nominal | | D104 Bernhard | | | D104 TuPS | | $w_i$ | $w_i$ |
|------|------|------|------|------|------|------|------|------|------|
|      | Mean | FWHM | Base | FWHM | Top | Mean | FWHM | AD | CD |
| A1 | 305.5 | 0.9 | 1.86 | 1.01 | 0.16 | 305.61 | 1.10 | 1 | 0 |
| C1 | 311.5 | 0.9 | 1.94 | 1.06 | 0.18 | 311.58 | 1.10 | 0 | 1 |
| D1 | 317.5 | 0.9 | 2.12 | 1.20 | 0.28 | 317.60 | 1.30 | 1 | 1 |
| A2 | 325.0 | 2.9 | 5.00 | 3.56 | 1.06 | 325.13 | 3.72 | -1 | 0 |
| C2 | 332.4 | 2.9 | 5.94 | 3.71 | 1.48 | 332.47 | 3.96 | 0 | -1 |
| D2 | 339.9 | 2.9 | 6.88 | 4.20 | 1.52 | 339.95 | 4.32 | -1 | -1 |



### 2.3.2 Brewer

Slit weighting functions for each Brewer instrument are derived from dispersion tests, which are typically part of the yearly
calibration. A detailed explanation of the calibration process, and the computation of the dispersion relation is given in Gröbner et al. (1998) and  Redondas et al. (2018). In short, the scanning mode in combination with the emission lines of different discharge lamps are used to determine the central wavelength and the FWHM of every slit by analysing the measured photon counts as a consequence of the illumination. In the standard operating procedure, the resulting triangle function of each slit is then truncated at 0.87 of the maximum height, and thus parametrized as trapezoids. The results of the calibration process are
typically given in a file ("lf-file"), and are summarized in Table 2 for both Brewer instruments. Brewer slit functions are instrument specific and can also vary over time. Redondas et al. (2018), using Brewer slit functions measured by a tuneable laser system similar to Köhler et al. (2018), report changes in the effective ozone absorption coefficients of the order of 0.8%. This is similar to the magnitude of changes we find for different Dobson slit measurements or parametrizations (Köhler et al., 2018).

**Table 2: Central wavelength (mean, nm) and full width at half maximum (FWHM, nm) of the individual slit functions for the Brewer instruments**

| Slit | B010 | | B226 | | $w_i$ |
|---|---|---|---|---|---|
| | Mean | FWHM | Mean | FWHM | |
| 2 | 306.308 | 0.520 | 306.275 | 0.527 | 0 |
| 3 | 310.055 | 0.514 | 310.026 | 0.520 | 1 |
| 4 | 313.505 | 0.538 | 313.471 | 0.528 | -0.5 |
| 5 | 316.809 | 0.528 | 316.778 | 0.522 | -2.2 |
| 6 | 320.013 | 0.520 | 319.963 | 0.512 | 1.7 |

## 2.4  Ozone absorption cross sections

The operational TOC retrieval for Brewer and Dobson instruments relies on the ozone absorption cross section measured by Bass and Paur (1985, B&P). As mentioned, several studies (Fragkos et al., 2015; Gröbner et al., 2021; Orphal et al., 2016;
Redondas et al., 2014) suggest using updated ozone absorption cross sections. This study focuses on four ozone absorption cross sections, all of which cover the wavelength range of 300 nm to 345 nm for the Brewer and Dobson spectrometers. Additionally, only datasets providing a quadratic polynomial approximation for the $T_{eff}$ dependency of the cross sections were considered.

- **SG14**. This dataset (Gorshelev et al., 2014; Serdyuchenko et al., 2014) comes from the Institute of Environmental
Physics at the University of Bremen. It provides data in the spectral range of 213 – 1100 nm with a spectral resolution of 0.02 – 0.24 nm. Temperature sensitivity was measured at 10 K intervals between 193 K and 293 K. Here, we use





the            dataset            downloaded            from            https://www.iup.uni-bremen.de/gruppen/molspec/databases/referencespectra/o3spectra2011/index.html. According to the authors, the dataset's uncertainty is 2 to 3%, depending on the wavelength region. This is consistent with other broadband cross-section measurements. Recent studies (Gröbner et al., 2021; Orphal et al., 2016; Redondas et al., 2014) have recommended this dataset, which minimizes the discrepancy between Dobson and Brewer measurements. Note that these studies referred to the dataset as "IUP" or "SER".

- **SG16**. This dataset (Weber et al., 2016) is very similar to the SG14 dataset, but additionally it provides detailed wavelength dependent uncertainty information, based on Monte Carlo simulations, and including uncertainties from the temperature parametrization as well as uncertainties from the laboratory measurements. The dataset was obtained from https://www.iup.uni-bremen.de/UVSAT/datasets/uv-ozone-absorption-cross-sections. The authors estimated an uncertainty of 1.1 to 3 %, depending on wavelength. In the Huggins band, for example, the overall uncertainty was estimated to be 1.5 % (1 σ).

- **G17**. This dataset (Gorshelev et al., 2017) is also linked to the above mentioned datasets. It was created as part of the ATMOZ ("Traceability for Atmospheric Total Column Ozone") Joint Research Program (JRP) funded by EMRP. It covers the wavelength range 295 – 350 nm, and is available for 11 temperatures between 193 and 293 K. The polynomial quadratic equation is not publicly available, but was provided via personal communication (Mark Weber, personal communication, 2023). Currently, no peer-reviewed publication with comprehensive details is available. However, the authors mention (Gorshelev et al., 2017) that the combined uncertainties are below 1%, and only increase near the spectral boundaries of the measurements. This dataset is similar to the dataset referred to as "IUP_A" in Gröbner et al. (2021), albeit has updated polynomial coefficients for the temperature dependency.

- **BW**. The dataset (Birk and Wagner, 2021) was measured in the framework of ESA project SEOM-IAS at the German Aerospace Center, for the wavelength region 243 – 346 nm, and at 6 temperatures in the range 193 – 293 K. Their polynomial   temperature   parametrization   was   downloaded   from   the   Zenodo   repository https://zenodo.org/record/4423918#.ZCFXQfbP1aT. Notably, we use the "version 2" dataset from the Zenodo repository. Currently, no peer-reviewed publication containing all details is available. This dataset is similar to the dataset referred to as "ACS" in Gröbner et al. (2021), who, however, determined their own polynomial temperature dependence.

Generally, the temperature dependence of all four new ozone cross sections uses a quadratic polynomial (see also Bass and Paur, 1985; Weber et al., 2016):

$$\sigma_T = C_0 + C_1 T + C_2 T^2 \tag{Eq. 9}$$





where $C_0$, $C_1$, and $C_2$ are the temperature coefficients (provided in the datasets). Fig. 3 illustrates the influence of various temperatures on the ozone absorption cross-sections for the SG16 dataset (blue lines). Differences between the B&P and SG16 cross-sections are shown by the dashed red line in Fig. 3. Below 320 nm, these differences are quite small. Above 325 nm, however, they become larger and often exceed several percent. Similarly, from the differences between the various blue lines, one can see that temperature effects are generally much larger at wavelengths longer than about 330nm.

Looking also at the slit-weighting functions for Brewer (dark grey, below 320 nm) and Dobson (light grey, also above 320 nm) in Fig. 3, one can already expect that both the change from B&P to SG16 cross-sections, and the application of temperature-dependent ozone cross-section, will have a much larger effect for the Dobson data, and only a small effect for the Brewer data.

    It is also worth noting that the vacuum wavelengths have to be converted to wavelengths in air. To do so, we utilized a python

script from Github (https://github.com/polyanskiy/refractiveindex.info-scripts/blob/master/scripts/Ciddor%201996%20-%20air.py), which employs the equation proposed by Ciddor (1996).

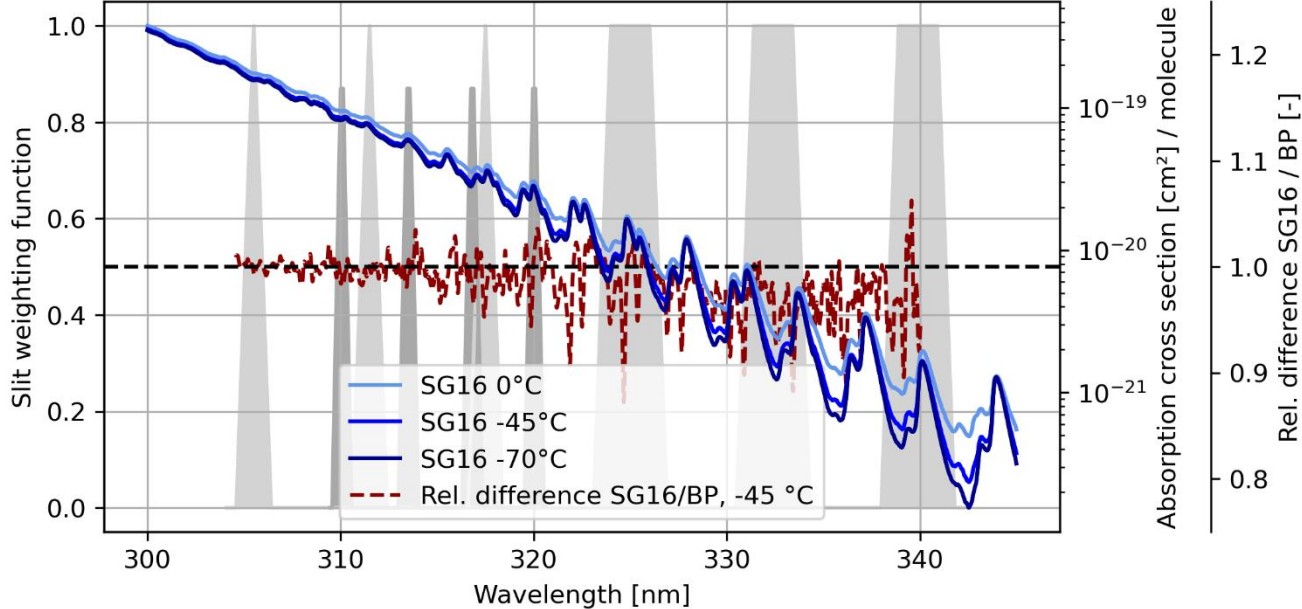


**Figure 3: Ozone absorption cross sections based on the SG16 dataset for three selected temperatures (blue curves). The red line shows the relative difference between the SG16 and B&P datasets for a temperature of -45 °C. The Dobson (light grey) and Brewer (dark grey) slit functions are shown as well.**





## 2.5 Effective ozone temperature

We follow other studies (Gröbner et al., 2021; Redondas et al., 2014; Scarnato et al., 2009; Vanicek, 2006) and use the effective ozone temperature $T_{eff}$ to describe the temperature effect of the ozone absorption cross sections. $T_{eff}$ can be computed from vertical profiles of temperature $T(z)$ and ozone density $O_3(z)$ based on the following equation:

$$\boldsymbol{T_{eff}} = \frac{\int T(z)O_3(z)dz}{\int O_3(z)dz} \qquad \textbf{(Eq. 10)}$$

Generally, $T_{eff}$ can be derived from modelled data, or from measurements. In our case, we compared two different $T_{eff}$ datasets to check whether the two approaches have significant differences:

- The TEMIS dataset contains $T_{eff}$ values produced by the European Centre for Medium-Range Weather Forecast (ECMWF). We downloaded station overpass files for Hohenpeissenberg and other locations from the Tropospheric Emission Monitoring Internet Service (TEMIS) website at https://www.temis.nl/climate/efftemp/overpass.php.

- OS_LIDAR dataset. This dataset combines ozone sonde measurements (altitude < 29 km) with LIDAR measurements (altitude ≥ 29 km) from Hohenpeissenberg.

Missing LIDAR or ozone sonde observations were filled using linear interpolation between available measurements. To ensure correct $T_{eff}$ calculations, it is mandatory to use vertical profiles of T and $O_3$ from the ground up to about 50 km altitude, where the $O_3$ density approaches zero. For the case of Hohenpeissenberg, using only ozone sonde data, which only reach burst heights of approximately 30 - 35 km, would result in a low bias of about 2.2°C (green dotted lines in Fig. 4).

Figure 4 depicts the temporal evolution of $T_{eff}$ over a two-year period, along with the 30-year $T_{eff}$ climatology (1990-2020) for both datasets (TEMIS and OS_LIDAR) at the Hohenpeissenberg site. The figure demonstrates the small differences between the two datasets and the sometimes-larger differences between daily and climatological values. While the climatological difference between TEMIS and OS_LIDAR is almost negligible, there can be differences up to ±2.5 °C between daily Teff data from the two sources (grey line in Fig. 4, bottom panel). However, in general, the two datasets are very similar, with a mean difference of approximately 0.1 °C and a standard deviation of about 1.2 °C for daily values in the 1990-2020 timeframe.

Larger differences occur between daily and climatological values (orange line in the bottom panel of Fig. 4). Especially in winter, the difference between daily and climatological values can reach ±8K. Overall the differences are less than a few K, and have a standard deviation of 2.2 °C for the TEMIS dataset. This is comparable to the size of differences between the

TEMIS and OS_LIDAR datasets.

In summary, Fig. 4 indicates that the use of climatological values for Teff already provides a very good representation of temperature variations over the year. In summer, very little can be gained by using daily values. Even in winter, differences between daily and climatological values are of similar magnitude as differences between the TEMIS and OS_LIDAR datasets. These findings bear significant relevance for selecting an appropriate dataset for operational or reprocessing purposes.





**Figure 4: Timeseries of ozone effective temperature T_eff based on TEMIS, on ozone sonde data (OS_only), or on combined ozone sonde (OS) and LIDAR data (top panel). The climatological values (climate, orange dashed lines) are calculated from daily values over the time period 1990 – 2020. Additionally, a 7-day rolling mean is applied. The bottom panel shows the daily difference between the TEMIS and OS/OS_LIDAR derived datasets.**

A look at the seasonal variation of $T_{eff}$ in other locations worldwide is presented in Fig. 5. Generally, stations at higher latitudes have a higher amplitude of the seasonal $T_{eff}$ cycle. In addition, higher latitudes also see much higher variability of $T_{eff}$, especially during winter and spring, as clearly shown by the shaded regions in Fig. 5.



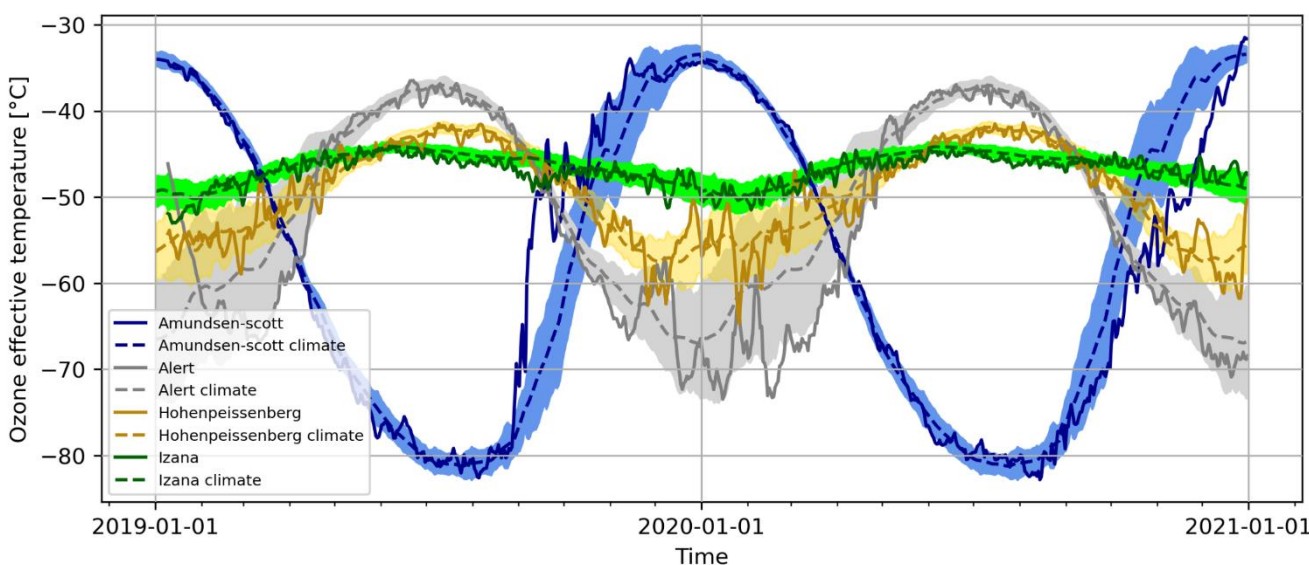

**Figure 5: Timeseries of TEMIS-derived ozone effective temperature $T_{eff}$ for 4 locations covering latitudes from -90° (Amundsen-Scott) to +82.5° (Alert). The dashed lines indicate the long-term climatology (1990-2020), and the shaded areas indicate the year to year variability (1 σ).**

### 2.6 Ozone absorption coefficients

We use the standard approach based on Komhyr et al. (1993) to calculate the differential ozone absorption coefficients Δα for Brewer and Dobson instruments. The approach involves using the effective ozone temperature $T_{eff}$, the polynomial temperature approximation for the ozone absorption cross sections $\sigma(\lambda, T_{eff})$, and the slit weighing functions $S_i(\lambda)$ for slit $i$, to calculate $\alpha_i$. This approach was used and is discussed in detail in multiple studies (Bernhard et al., 2005; Gröbner et al., 2021; Redondas et al., 2014, 2018). It is defined by the following equation:

$$\alpha_i = \frac{\int \sigma(\lambda, Teff) S_i(\lambda) d\lambda}{\int S_i(\lambda) d\lambda} \qquad \text{(Eq. 11)}$$

Applying the polynomial expression for the ozone cross sections (Eq. 9) and rearranging the equation provides a polynomial equation for αi, using $T_{eff}$ and a set of coefficients Aij:

$$\alpha_i(T_{eff}) = A_{i0} + A_{i1} * T_{eff} + A_{i2} * T_{eff}^2 \qquad \text{(Eq. 12)}$$

with



$$A_{ij} = \frac{\int C_j(\lambda) S_i(\lambda) d\lambda}{\int S_i(\lambda) d\lambda}$$

(Eq. 13)

where the $C_j(\lambda)$ are the coefficients for temperature dependence from Eq. (9), and the $S_i(\lambda)$ are the slit functions. The resulting
$A_{ij}$ coefficients for individual slits of the Dobson instrument, based on the SG16 ozone absorption cross section dataset are
listed in Table 3. The combined coefficients required for a Brewer or Dobson TOC measurement (see Eq. 5), e.g., for the AD-
wavelength pair (AD = A – D), are obtained by summing up the individual, slit-dependent coefficients, with their
corresponding weights following Eq. (8).

Note that the coefficients in Table 3 are very similar to those published by Redondas et al. (2014), which, to our knowledge,
is the only reviewed publication that directly reported the coefficients utilizing the new ozone cross sections.

**Table 3: Coefficients for the temperature dependence (in °C) of the effective ozone absorption cross section for the different Dobson slits (A, C, D). Results are based on the SG16 / SG14 dataset, and the slit approximation from Bernhard et al. (2005). For values of the ozone absorption coefficient at the currently fixed $T_{eff}$ see Table 5.**

| Dobson slit [nm] | Slit | Coef. A0 | Coef. A1 | Coef. A2 |
|---|---|---|---|---|
| 305.50 | A1 | 2.0622 | 4.4327e-03 | 2.0565e-05 |
| 325.00 | A2 | 1.3888e-01 | 7.0187e-04 | 3.5059e-06 |
| 311.50 | C1 | 9.5124e-01 | 2.6806e-03 | 1.3161e-05 |
| 332.40 | C2 | 4.9357e-02 | 3.0492e-04 | 1.6500e-06 |
| 317.50 | D1 | 4.2439e-01 | 1.4114e-03 | 7.3122e-06 |
| 339.90 | D2 | 1.4984e-02 | 1.2597e-04 | 6.6166e-07 |
| **AD** | | **1.5139** | **2.4454e-3** | **1.0409e-5** |
| **CD** | | **0.4925** | **1.0903e-3** | **4.8607e-6** |
| AD* | | 1.5157 | 2.4502e-03 | 1.0518e-05 |
| AD** | | 1.5133 | 2.4403e-3 | 1.0356e-5 |
| CD** | | 0.4926 | 1.0924e-3 | 4.8841e-6 |

* data from Redondas et al. (2014)

** Independent evaluation of co-author Julian Gröbner

Table 4 summarizes the resulting coefficients for temperature dependence of the combined differential ozone absorption
coefficients. Results are shown for different ozone cross section data sets and for Dobson104 and Brewer010. Due to their
potentially different instrument specific slit functions, other Brewers will have slightly different coefficients. Based on the
mean of 123 dispersion tests of 33 Brewer instruments, for example, Redondas et al. (2014) calculated coefficients using the
SG14 data set (A0=3.4591e$^{-01}$, A1=2.8781e$^{-5}$, A2=-4.9188e$^{-8}$), comparable to ours, and giving an instrument-specific $\Delta\alpha$ within
0.06 % from our Brewer010 value. Note also the much smaller temperature dependence for the Brewer where A1 and A2 are
about two orders of magnitude smaller than for the Dobson.





**Table 4: Coefficients for the temperature-dependence of the combined ozone absorption coefficient for the main Dobson wavelength pairs (AD, CD), and for the Brewer010. For the Dobson instrument, the slit approximation from Bernhard et al. (2005) was applied. Results are shown for the different ozone cross section datasets. SG14 and SG16 are combined because results are very similar.**

| | Dobson104 | | | | | | Brewer010 | | |
| | AD wavelength pair | | | CD wavelength pair | | | | | |
| | SG14/SG16 | G17 | BW | SG14/SG16 | G17 | BW | SG14/SG16 | G17 | BW |
|---|---|---|---|---|---|---|---|---|---|
| A0 | **1.5139** | 1.5182 | 1.6328 | **4.9247e-01** | 4.8846e-01 | 5.5360e-01 | 3.4555e-01 | 3.4685e-01 | 5.0591e-01 |
| * | 1.5157 | | | | | | | | |
| ** | 1.5133 | | | 4.9259e-01 | | | | | |
| A1 | **2.4453e-03** | 2.5650e-03 | -3.5454e-03 | **1.0903e-03** | 9.6121e-04 | -1.4113e-03 | 1.9485e-05 | 9.5578e-05 | -1.2244e-03 |
| * | 2.4502e-03 | | | | | | | | |
| ** | 2.4403e-03 | | | 1.0924e-03 | | | | | |
| A2 | **1.0409e-05** | 1.0682e-05 | 1.1250e-05 | **4.8607e-06** | 3.7791e-06 | 4.2883e-06 | -1.7734e-07 | 1.3213e-06 | 2.4152e-06 |
| * | 1.0518e-05 | | | | | | | | |
| ** | 1.0356e-05 | | | 4.8841e-6 | | | | | |
| $\alpha_{(op-Teff)}$ | **1.4230** | 1.4223 | 1.4074 | **0.4524** | 0.4521 | 0.4541 | 0.3443 | 0.3452 | 0.3523 |

* data from Redondas et al. (2014)

** Independent evaluation of co-author Julian Gröbner

Finally, Table 5 gives a comparison of the effective differential ozone absorption coefficients at the currently used fixed temperatures for Dobson and Brewer, for the different cross section data sets, and different slit functions. For the Dobson, all
new cross section data sets give 0.6% to 2.2% smaller effective ozone absorption coefficients than B&P. This would result in correspondingly larger total ozone values. BW stands out with the smallest effective absorption coefficient. These results are very similar to Gröbner et al. (2021) and Redondas et al. (2014), which are also shown in the table. Note, however, the slightly smaller effective cross sections, about 0.6% smaller, for Dobson104 for the handbook´s slit functions, compared to Bernhard or TuPS.

For the Brewers, all new cross section data sets give 0.9% to 3.3% larger effective ozone absorption coefficients than B&P. This would result in correspondingly smaller total ozone values. Again, the BW dataset stands out with the largest Δα. As mentioned, Brewers have different slit functions for different instruments. Here this results in about 2% larger Δα for Brewer 226 compared to Brewer 10. Both are within the range of Δα reported by Redondas et al. (2014). Based on 33 Brewer instruments and 123 dispersion tests, they report values between 0.335 cm$^{-1}$ and 0.350 cm$^{-1}$ for both the B&P and the SG14
dataset.





**Table 5: Effective differential ozone absorption coefficient (in atm cm$^{-1}$) at the nominal fixed T$_{eff}$ for the different ozone cross section datasets, for Dobson and Brewer instruments. The Dobson results are given for three different slit functions. The fixed operational T$_{eff}$ is -46.3 °C for Dobsons, and -45 °C for Brewers.**

| Slit definition / Ozone absorption cross section | Dobson 104, default T$_{eff}$ = -46.3 °C | | | Brewer, default T$_{eff}$ = -45 °C | |
|---|---|---|---|---|---|
| | Handbook | Bernhard | TuPS | Brewer010 | Brewer226 |
| | $\alpha_{AD}$ / $\alpha_{CD}$ | $\alpha_{AD}$ / $\alpha_{CD}$ | $\alpha_{AD}$ / $\alpha_{CD}$ | | |
| B&P | 1.432 / 0.459 | | | 0.3411 | 0.3484 |
| SG14 | 1.4148 / 0.4491 | 1.4229 / 0.4525 | 1.4232 / 0.4446 | 0.3445 | 0.3517 |
| SG16 | 1.4149 / 0.4490 | **1.4230 / 0.4524** | 1.4231 / 0.4446 | 0.3443 | 0.3516 |
| G17 | 1.4141 / 0.4487 | 1.4223 / 0.4521 | 1.4226 / 0.4440 | 0.3452 | 0.3524 |
| BW | 1.4012 / 0.4523 | 1.4074 / 0.4541 | 1.4050 / 0.4459 | 0.3523 | 0.3550 |
| SG14* | | 1.4250 / ------- | | 0.333 to 0.350 | |
| SG14** | | 1.425 / ------- | 1.429 / ------ | | |
| SG14*** | | 1.4225 / 0.4523 | | | |

\* data from Redondas et al. (2014), for Brewer based on 33 instruments and 123 dispersion tests.

\*\* data from Gröbner et al. (2021)

\*\*\* Independent evaluation of co-author Julian Gröbner

When the new effective differential ozone cross sections are known, the relationship between TOC and Δα in Eq. (14) allows for easy reprocessing of TOC values. Currently, the differential ozone absorption coefficient Δα$_{OP}$ is based on the B&P cross sections at a fixed temperature and the Komhyr parametrization. By applying the following equation, the corresponding old operational TOC values can easily be recalculated to the new ozone cross sections Δα$_{Teff}$ with varying T$_{eff}$:

$$TOC_{Teff} = TOC_{OP} \frac{\Delta\alpha_{OP}}{\Delta\alpha_{Teff}}$$

(Eq. 14)

As mentioned, knowledge about the slit functions is necessary. This is easier for the Dobson instrument, as the slit functions are wider and generally quite similar for all Dobson instruments (Köhler et al., 2018). However, for the Brewers, the slit functions are narrower and are typically determined individually for each instrument from dispersion tests during calibration campaigns. Therefore, for Brewers, the history of parameters that describe the instrument-dependent slit functions (e.g., central wavelength, FWHM) must be available for the most accurate recalculation. Nevertheless, Redondas et al., (2014) also demonstrated that historical ozone measurements from Brewer instruments can be effectively corrected, with a TOC error of less than 0.2%, by employing a linear relationship dependent only on the central wavelength of the respective Brewer instrument, while disregarding the shape of the slits.





# 3    Results and discussion

## 3.1    Temperature dependency of Δα

Figure 6 shows our results for the temperature-dependent effective absorption coefficients $\Delta\alpha_{Teff}$ for the various instruments and ozone cross section datasets. While the standard operating procedure for the Brewer and Dobson instruments uses fixed effective ozone temperature (-45°C and -46.3°C, respectively), the real Δα varies strongly with $T_{eff}$, as shown in Fig. 6. The figure also shows the much larger impact of $T_{eff}$ on the effective absorption coefficient for the Dobson, ranging from -3% to +3% in the top panel of Figure 6, compared to the smaller effect for the Brewer, ranging from -0.5% to +2%. While the

different ozone cross sections (G17, SG14, SG16, BW) have only a very minor impact on the temperature dependence of $\Delta\alpha_{Dobson}$, they clearly result in very different temperature dependencies for $\Delta\alpha_{Brewer}$, especially for the lower range of $T_{eff}$. In addition, the instrument specific slit functions play a role for the Brewer, as can be seen in the slight differences between the results for BR010 and BR226 in the bottom panel of Fig. 6.

Looking at the temperature dependence of the absorption coefficient Δα in Fig. 6, and at the variations of $T_{eff}$ in Fig. 5, it

becomes quite obvious that implementation of temperature-dependent ozone-cross sections in the operational retrieval algorithm is important, especially for the Dobson. It should reduce the uncertainty of Dobson TOC values by several percent, while improvements for Brewers will generally be smaller (and limited e.g. by the knowledge of the slit functions of the individual instruments).




**Figure 6: Temperature dependence of the ozone absorption coefficients for different ozone cross section datasets (SG14, SG16, G17, BW) for Dobson (D104, top panel) and Brewer (bottom panel, B010: solid lines; B226: dashed lines) instruments. The dependence is calculated relative to the respective absorption coefficient for an effective temperature of -45 °C. For the Dobson, only the results of the AD wavelength pair are shown.**






## 3.2 Updated Brewer and Dobson TOC values

The relative difference in TOC when using either the B&P operational ozone absorption coefficients, or the new $T_{eff}$-dependent absorption coefficients can be calculated using Eq. (14). Table 6 shows the resulting average TOC changes. Generally, the use
of the new ozone cross sections leads to increased Dobson TOC values (by 0.9 to 2.5%), and to decreased Brewer TOC values (by -1 to -3%). The BW dataset provides by far the largest changes, both for Brewer and Dobson. It would increase the differences between Dobson and Brewer, and appears not to be suitable. The three remaining datasets provide mean TOC changes in the range of 0.9 – 1.6 % for the Dobson instrument, and -1.2 – -0.9 % for the Brewer instrument. Generally, the SG14 and SG16 datasets, along with the Bernhard slit approximation or the TuPS measurement for the Dobson instrument,
exhibit the smallest differences compared to the B&P operational dataset.

**Table 6: Mean [%] and standard deviation [1σ, %] of the relative difference in TOC between four $T_{eff}$-dependent ozone absorption cross sections and the operational B&P dataset with a fixed $T_{eff}$, for both the Brewer and Dobson instruments and at Hohenpeissenberg. The differences were calculated using climatological TEMIS $T_{eff}$ data (1990 – 2020), and the values show the**
**averaged results for a period of one year. The results also correspond to the dashed grey lines in Fig. 7 for the location of Hohenpeissenberg.**

| | Dobson 104 | | | Brewer | |
|---|---|---|---|---|---|
| $TOC_{new}/TOC_{B\&P}$ | Handbook | Bernhard | TuPS | Brewer010 | Brewer226 |
| | mean ± std | mean ± std | mean ± std | mean ± std | mean ± std |
| SG14 | 1.5±0.6 | 1.0±0.5 | 0.9±0.5 | -0.9±0.1 | -0.9±0.0 |
| SG16 | 1.5±0.6 | 0.9±0.5 | 0.9±0.5 | -0.9±0.1 | -0.9±0.0 |
| G17 | 1.6±0.6 | 1.0±0.6 | 1.0±0.5 | -1.2±0.1 | -1.2±0.1 |
| BW | 2.5±0.6 | 2.1±0.6 | 2.2±0.5 | -3.4±0.2 | -2.1±0.2 |

Fig. 7 displays corresponding time series for the differences in TOC between the operational B&P derived values and the SG16 dataset over a period of two years, and for four different stations from -89.98° to 82.45°. Generally, the new temperature-
dependent ozone absorption coefficients lead to larger changes in TOC values at higher latitudes (due to the higher variability in $T_{eff}$). In contrast, TOC values close to the tropics vary by less than 1% when the new ozone cross sections are applied. Similarly, the impact of using either climatological $T_{eff}$ values or daily values is much more pronounced for the Dobsons, especially at higher latitudes. Nevertheless, as shown in Fig. 7, for the majority of TOC measurements world-wide, the difference between Dobson TOC values obtained from climatological instead of daily $T_{eff}$ values will be less than 1% (2σ).
For the Brewers, temperature dependence is much smaller, and there is virtually no difference between using climatological or daily $T_{eff}$ values.



Figure 7: **Relative difference in TOC between new Teff dependent ozone cross sections (SG16, TEMIS climate) and fixed temperature B&P cross sections (grey dashed lines), and between daily and climatological values for Teff (colored shaded regions, SG16 cross section, Teff daily and climatology from TEMIS). Results are given for four locations and Dobson (left panels) and Brewer (right panels). The shaded areas show the potential difference in TOC (2 σ) when using climatological $T_{eff}$ (1990 – 2020) instead of daily TEMIS values. Bernhard slit approximation was used for the Dobson instrument. For the Brewer, the slit functions from Brewer010 as described in Table 2 were applied.**

## 3.3   Comparison of Brewer and Dobson TOC retrievals

Consistency between TOC measurements from Dobson and Brewer instruments is crucial for evaluating whether a new ozone cross section dataset is recommended in this study. Fig. 8 shows TOC measurements from Dobson104 compared to two Brewer instruments, for the different ozone cross-section datasets.

As already shown in Fig. 1, a seasonal variation is quite prominent for the B&P dataset without $T_{eff}$ -correction (blue lines). In contrast, the SG14/SG16 cross sections produce almost identical results (black lines) and reduce the seasonal variation to less





than ±0.5%. There is also little difference between the results for Brewer 010 and Brewer 226 (solid and dashed lines, respectively). The overall Dobson to Brewer difference is close to zero. The G17 dataset results in a slightly negative Brewer-Dobson average difference and also very small annual variation (red lines). Much larger mean differences are seen for the

B&P OP and BW datasets (blue and orange lines, respectively). While the annual variation is also small for the BW dataset, it also shows a very large difference between the two Brewers. On the basis of Fig. 8, it is quite clear that the SG14 and SG16 datasets provide the best overall agreement between Dobson and Brewer measurements.

Gröbner et al. (2021) also identified a large offset for the BW dataset, up to 2.1% using measured slit weighing functions for their Dobson instrument. Generally, our findings are very similar to those of the previous studies by Gröbner et al. (2021) and

Redondas et al. (2014) who, for their stations and instruments, found mean differences for the SG14 dataset in the range of 0 to 1 % and -0.4 to 0.2 %, respectively. Note that Gröbner et al. (2021) found a larger difference, -1.0 to -1.5%, when using their version of the G17 cross sections, whereas in our study the G17 dataset generally performed very well. Partly, this difference can be attributed to different Rayleigh coefficients applied (see Eq. 5). Gröbner et al. (2021) used Bodhaine's Rayleigh cross-section (Bodhaine et al., 1999), whereas we applied the Rayleigh cross sections from the standard Brewer and

Dobson algorithm (Bates, 1984; Komhyr and Evans, 2008). Gröbner et al. (2021) states that applying Bodhaines´s values in Davos decreases TOC from Dobson by about -0.5 DU, and TOC from Brewer by about -2.4 DU. This may contribute approximately -0.6 to -0.7% to the -1.0 to -1.5% difference found in Gröbner (2021). The rest seems to be due to an older version of the G17 dataset used by Gröbner et al. (2021) This ambiguity in the G17 dataset, and the lack of an official publication, leads to the overall recommendation to use the SG16 dataset, and not G17.

The choice of slit approximation for the Dobson instrument also influences the comparison. Generally, the best comparison is achieved with the Bernhard approximation or the TuPS measurements (see also Fig. S5 in the supplement). In our case both outperform the Dobson operations handbook's slit approximation. This is generally consistent with the findings of Gröbner et al. (2021), who, however, obtained slightly better results when using the TuPS measurements. Based on our experience, and many previous measurements, including a number of Dobson slit measurements world-wide (Gröbner et al., 2021; Köhler et

al., 2018), it seems that for a majority of Dobson instruments the Bernhard et al. (2005) slit approximation is indeed very good, is simple and is suitable for the entire network. Due to its ease of implementation, and the good results in our study, we recommend the Bernhard slit approximation for Dobson instruments in the operational network.

Where available, TuPS measurements of Dobson slit functions can result in small improvements (typically of the order of 0.5% or less), albeit at the cost of additional measurements, additional calculations and much more extensive housekeeping.

This may make sense for some specialized research groups, but for the wide network we feel that the Bernhard slit approximation is adequate and simple to keep track of.





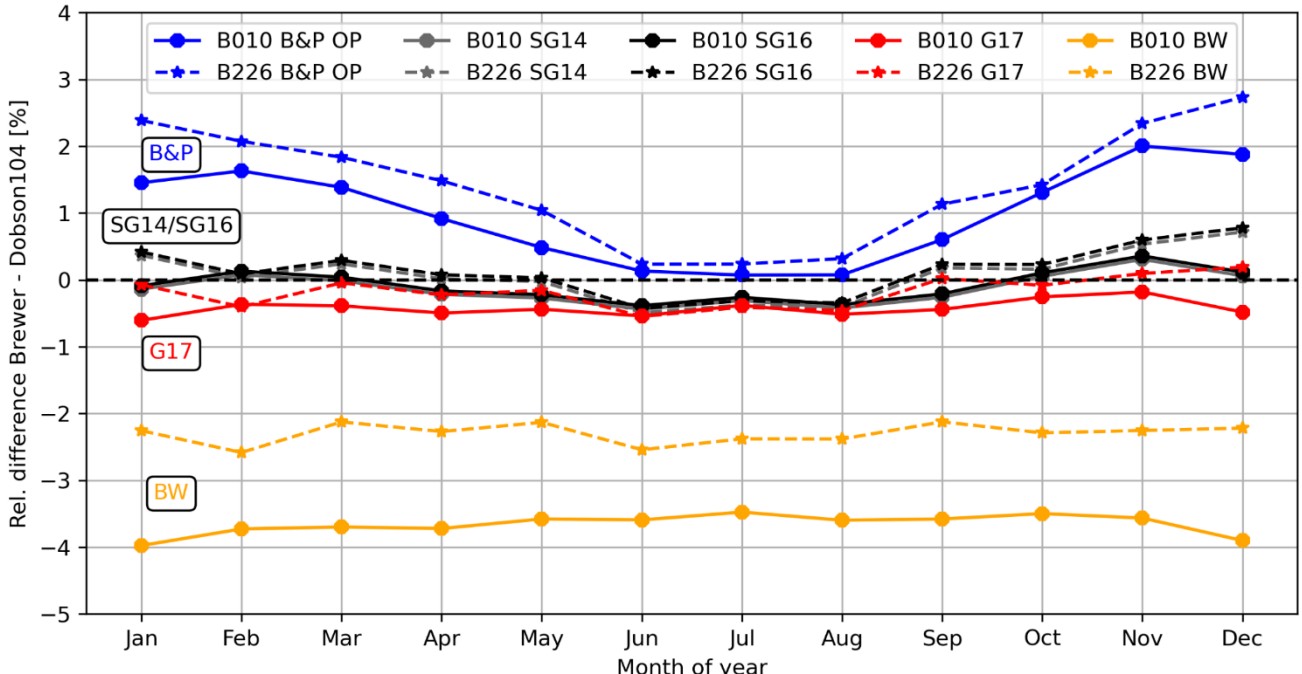

**Figure 8: Monthly mean difference between Dobson 104 and Brewers 010/226 (solid and dashed lines), at Hohenpeissenberg and using the Bernhard slit approximation. The different colors represent the results for the different ozone absorption cross section datasets.**

### 3.4 Uncertainties

A comprehensive analysis of uncertainties for Dobson total ozone measurements is given by Basher (1982), for Brewers information on uncertainty can be found in different publications (Fioletov et al., 2005; Kerr and McElroy, 1995; Redondas et al., 2018; Zhao et al., 2021). A new assessment of these uncertainties is beyond the scope of this paper. It is the topic of two separate papers in preparation by some of the co-authors. Nevertheless, it is important to consider the various sources of uncertainty here, and to determine when further reductions of uncertainty from a single source will not improve the overall uncertainty.

Basher (1982) separates in his analysis between typical good instruments or situations (with smaller uncertainties) and bad instruments or situations (with large uncertainties). Here, we will consider only the good case. In a similar way to Basher we separate between (1) instrumental sources of uncertainty (alignment, calibration, slit functions, instrumental noise, …), (2) uncertainty due to simplified radiative transport assumptions (aerosol and $SO_2$ interference, ozone layer height, airmass calculation, …), and (3) uncertainty due to the used ozone absorption cross-sections (3a) and their temperature dependence (3b). All of these uncertainties contain random and systematic parts.

For a typical "good" Dobson, the instrumental relative standard uncertainties (1) are estimated to be less than 0.5% to 1.5% by Basher (1982). This is consistent with the standard deviation / repeatability of individual Dobson TOC values observed at





Hohenpeissenberg, which is about 0.7%, and the typical agreement reached in Dobson calibrations, which is also about 0.7 %.
It is also consistent with the magnitude of changes due to different wavelengths or slit functions found in this study, which are
about 0.5%, as can be seen in Fig. S5 in the Supplement, and Table 7. Similar uncertainties of this type apply also for Brewers.
The standard deviation / repeatability of individual Brewer TOC values observed at Hohenpeissenberg, for example, is about
0.9%.

Relative standard Uncertainties (2), due to the simplified radiative transfer assumptions, are also of the order of 0.2% to 0.5%
for a "good situation" Dobson or Brewer. This study does not address any of these sources of uncertainty, so their values
remain unchanged.

The largest improvement coming from this study is in the application of new ozone absorption cross-sections with reduced
uncertainty (3a), and particularly in now addressing the temperature dependence ($T_{eff}$) of the ozone cross-sections (3b). Basher
quotes an absolute relative standard uncertainty due to the used ozone cross sections of about 3%, and a relative standard
uncertainty of about 1.5% due to neglecting the temperature dependence. The SG16 cross sections recommended here claim a
smaller absolute uncertainty (3a), about 1.5%, which would apply to both Dobson and Brewer TOC values. The major
improvement comes from addressing the temperature dependence (3b, $T_{eff}$), which reduces the associated uncertainty for
Dobson TOCs from about 1.5% (compare also the dashed lines in Fig. 7) to less than 0.5% (compare also the shaded regions
in Fig. 7). For Brewers, the uncertainties associated with Teff are much smaller, and are assumed to be about 0.1% (see also
right panels in Fig. 7, and Koukouli et al., 2016).

In summary, the improved processing suggested in this paper should reduce the combined relative standard uncertainty of
Dobson TOC values from 3.5% to 1.8%. For Brewer TOC values the improvement is smaller from 3.2% to 1.8% (Table 7).

**Table 7: Relative standard uncertainty estimation based on literature (Basher, 1982; Koukouli et al., 2016; Scarnato et al., 2009, 2010; Zhao et al., 2021) and our own study. Uncertainty sources 1,2,3a,3b correspond to the uncertainties associated to instrumental**
**sources (1), simplified radiative transport assumptions (2), applied cross sections (3a) and Teff (3b).**

| Uncertainty | Dobson | | Brewer | |
|---|---|---|---|---|
| | Operational | SG16, Teff-corr | Operational | SG16, Teff-corr |
| 1 | 0.7 | 0.7 | 0.9 | 0.9 |
| 2 | 0.5 | 0.5 | 0.5 | 0.5 |
| 3a | 3.0 | 1.5 | 3 | 1.5 |
| 3b | 1.5 | 0.5 | 0.1 | 0.1 |
| Combined | 3.5 | 1.8 | 3.2 | 1.8 |



### 3.5 Recommendations for operational networks

Based on our results, and taking into account previous studies (Gröbner et al., 2021; Köhler et al., 2018; Orphal et al., 2016;
Redondas et al., 2014, 2018), we recommend the SG16 ozone absorption cross sections for the Dobson and Brewer observing
networks. For the Dobson instruments, the slit approximation of Bernhard et al. (2005) should be applied. The correction for
the effective ozone temperature should be based on the TEMIS/ECMWF dataset.

- The SG16 (Weber et al., 2016) dataset performs very similar to the SG14 (Serdyuchenko et al., 2014) dataset, which
  was recommended in the previous studies. However, the SG16 dataset also provides uncertainty budgets, which is
useful for further studies.

- The G17 dataset (Gorshelev et al., 2017) provides similar results, but introduces a slightly larger mean difference
  between Dobson and Brewer, compared to SG16. Moreover, no peer-reviewed publication of the dataset exists at the
  time of this publication.

- The Bernhard slit approximation (Bernhard et al., 2005) outperforms the slit approximation of the Dobson Operations
Handbook (Komhyr and Evans, 2008), which introduces a small bias between Dobson and Brewer measurements.
  While some studies (Gröbner et al., 2021; Köhler et al., 2018) recommend instrument specific slit weighing functions
  (e.g. from TuPS measurements), the application of TuPS measurements did not result in improved consistency
  between Dobson and Brewer TOC measurements in this study. Moreover, only a very limited amount of reliable
  measured slit weighting functions from Dobson instruments exists to this day. This would delay and complicate the
implementation of new ozone cross sections in the operational networks quite a lot, without a large gain in the
  accuracy of the resulting TOC values.

- Slightly better results may be achieved, at considerable housekeeping cost, by utilizing measured slit functions (e.g.
  by TuPS) for the effective ozone absorption coefficients for Dobson instruments. Here it is crucial to ensure that the
  wings of the slit functions are included, particularly at the shorter wavelengths, where the ozone cross-sections are
large. While this may be the way to go for a few specialized research groups, for much of the operational network the
  simple and easily applied Bernhard slit approximation seems good enough and is therefore recommended.

- The TEMIS/ECMWF ozone effective temperature dataset is very well suited for application in the global Brewer and
  Dobson networks (https://www.temis.nl/climate/efftemp/overpass.php). Differences to measured values from a
  combination of LIDAR and ozone sondes are small, with a standard deviation of only about 1.2 °C for daily values
over a time period of 30 years. Climatological values derived from the dataset are sufficient for use in the operational
  networks. Daily data would not improve the quality of the TOC measurement significantly: negligible differences in
  the yearly mean, for the majority of observing stations differences smaller than 1 % for daily TOC data, larger
  differences only at high latitudes in winter, where measurements are problematic anyways due to low solar elevation.





The transition to new Dobson TOC values based on SG16 and the TEMIS $T_{eff}$ climatology should be carried out in a centralized facility, such as the WOUDC. Figure 9 outlines our suggested approach, which would make sure that all critical computations are applied uniformly to both existing historical and incoming new Dobson data. The central processing would use the TEMIS $T_{eff}$ climatology to calculate new effective absorption coefficients for each reporting measurement location. In addition, the central processing would ensure proper metadata handling (e.g. versioning, applied $T_{eff}$, polynomial function coefficients used

for B&P to SG16 conversion, …).

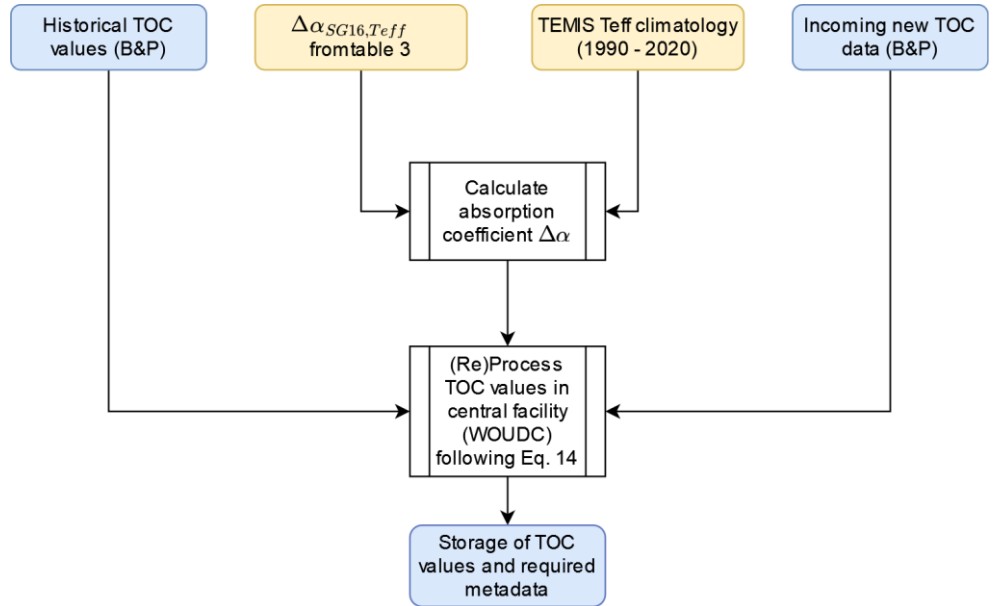

**Figure 9: Flowchart illustrating the suggested centralized transition to revised Dobson Total Ozone Column (TOC) timeseries in the**
**operational network, using the new $T_{eff}$ -dependent SG16 ozone absorption cross sections.**

## 4    Conclusions

Focusing on Dobson and Brewer total ozone measurements, this study reinvestigated the use of different ozone absorption cross section data sets, and different ways to account for ozone effective temperatures $T_{eff}$.

Overall, the SG16 ozone cross sections give the most consistent results. Therefore, it is recommended to implement the SG16

cross section in both the Brewer and Dobson networks. This will provide more consistent and accurate total ozone data.

For effective ozone temperature ($T_{eff}$) simply TEMIS climatological values yields satisfactory results for nearly all reporting stations. At most stations, very little can be gained by using daily $T_{eff}$ values.



Overall, the uncertainty of total ozone data from Dobson should improve from currently 3 to 4 % (due to 1 to 3 % annual variation in bias) to better than 2 % in the future. Much less can be gained for Brewer total ozone data, where the new cross-sections and $T_{eff}$ data only result in changes of the order of $\pm0.5\%$.

**Data availability.** The datasets used in this study will be made available through Zenodo at the final stage of the review process.

**Author contributions.**

KV processed and analyzed the datasets and authored the manuscript. WS conceptualized the study, offered valuable insights, and enhanced the manuscript's text. VV oversaw the Dobson and Brewer instruments at Hohenpeissenberg and contributed ideas and text enhancements. LE and JG provided their own experimental data, contributed ideas, and assisted in refining the paper. AR offered valuable insights and discussions, as well as substantial improvements to the manuscript during its final stages.

**Competing interests.** The authors declare that they have no conflict of interest.

**Acknowledgements.** We gratefully acknowledge funding from DWD (IAFE project Ozon_2025). We further want to thank Martin Adelwart, Michel Heinen and Marco Kirchner for taking nearly all the operational measurements and for their excellent support during the calibration campaigns. Finally, we would like to thank chatGPT (https://chat.openai.com, last accessed: 18 October 2023) for their assistance in improving the text on a few occasions.

**Financial support.**

**Review statement.**

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
