# Peer review of "The transition to new ozone absorption cross-sections for Dobson and Brewer total ozone measurements"

_Atmospheric Measurement Techniques, 2023_

## Author Comment (AC1)

amt-2023-220

**Author's response to comments**

We'd like to thank all reviewers for their very positive reviews, and for their valuable comments.

In the following, referees' comments are printed in black. Author responses are printed in blue. Here we list only the more major comments, which required a specific response. For all remaining minor (language related or technical) comments we have implemented the referees' suggestions.

**Important, but in the end minor, overall changes:**

Following the comment by Germar Bernhard about the erroneous value for the top width of the Dobson A2 slit function (our Fig. 2b), we have recalculated and replotted everything with the corrected top width for the Dobson A2 slit function (2.12 nm, not 1.06 nm). This did result in minor numerical changes to our previous results (typically less than 0.05%), well within the uncertainties mentioned throughout the paper. This recalculation provided additional confirmation for these uncertainties: Small changes less than 1% do occur, when assumptions or numerical inputs for slit functions, effective temperatures, etc. are changed.

Following up on a comment by reviewer #2, and after a conversation with Manfred Birk, we have also recalculated the results for the BW dataset. In the previous version of the manuscript, we did not convert the wavelengths from vacuum to air (see detailed response to reviewer#2). Applying this conversion resulted in changes of the BW TOC results by about 1%. It also resulted in better agreement between Dobson and Brewer (from 2-3 % difference to 1-2 %). Nevertheless, SG14 and SG16 still work better than BW. Thus, the conclusions of the manuscript remain unchanged.

**Author's response to comments of referee #1**

https://doi.org/10.5194/amt-2023-220-RC1

**Reviewer #1 general comments**

1. Although the paper is well written and needs no major corrections, there could be some confusion in the use of the terms "weighting coefficient", "slit weighting function" and "slit function". It may be better if the weighting coefficients are described more fully as the coefficients of the linear combination, and the difference, if any, between slit function and slit weighting function is explained. Otherwise if none, use one or the other. In tables 1 and 2 the weighting coefficients, w, are included. Again, this could be confusing as the discussion in that section is about the slit functions. It could be helpful to remind again here that these are then used to calculate the differential alpha.

We thank the reviewer for pointing this out. We fully agree. We have revised the text accordingly. The coefficients, 'w,' are now called coefficients of the linear combination. The term 'slit function' is now used consistently throughout the manuscript. The caption of Tables 1 and 2 was modified to clarify the meaning of the coefficients, 'w'.

**Author's response to comments of referee #2**

https://doi.org/10.5194/amt-2023-220-RC3

**Reviewer #2 general comments**

Line 267: A bit more information on the TEMIS dataset should be provided. Which ECMWF reanalysis has been used? ERA-5, ERA-interim, or something else? Which pressure levels and/or time stamps (e.g., daily mean or sunlit period only?) have been used in the Teff integration? The link provided did not show any further details. Or, at least, some references should be provided.

TEMIS data prior to 2000 rely on ERA-40, while data from 2000 onwards are based on the operational ECMWF analysis (personal communication, R. van der A, 2024). Further information on the dataset can also be found in Van der A et al. (2010), which is in the list of references.

We included this information in the manuscript.

Line 411: Could you please provide some reasonings why the BW dataset had the largest changes? Theoretically, good ozone xs will improve the agreement between Brewer and Dobson, so I fully agree with the authors that BW's performance is not ideal. But, some reasoning and/or explanations might be needed.

Many thanks for this very valuable comment. During reinvestigation of the ozone cross sections due to your comment, in particular the BW dataset, we identified an error in processing the BW data. We assumed, based on better agreement with Dobson ozone absorption coefficients from other cross sections (SG14, SG16, BP), that BW wavelengths didn't require conversion from vacuum to air. However, we now recognized, after a detailed comparison with the SG16 dataset (see Figure 1 below), and a personal communication with Manfred Birk from the German DLR (producer of the BW dataset), the necessity to perform the conversion for BW wavelengths as well. Unfortunately, the dataset does not explicitly state whether this conversion is required (it is not clearly defined whether vacuum or air wavelengths are given for the polynomial coefficients).

We now convert the BW wavelengths to air (similar to SG14 and SG16), and have recalculated and thoroughly examined the results. Manuscript and all figures were updated. The wavelength conversion resulted in slightly smaller ozone absorption coefficients for the Dobson instrument (and consequently higher TOC values), but also led to a better agreement with the Brewer instrument. In summary, the Brewer-Dobson agreement improved from previously 2-3% to now 1-2% (see Fig. 2 below).

However, despite these improvements, the results still fall short when compared to SG14/SG16. This leaves the manuscript's conclusions and main results unchanged.

For the BW cross sections, there appears to be an offset in the lower range of the wavelength region (<315nm, see Fig. 1). This is relevant for the absorption coefficients of the short wavelengths (A short, C short) for the Dobson instrument. The absorption coefficient based on SG16 for e.g. the short wavelength of the A pair is significantly larger compared to BW (1.9011 vs. 1.8652 atm cm-1, for $T_{eff}$ of -46.3 °C), which almost completely explains the differences in TOC from Dobson between BW and SG16. Nevertheless, addressing the question about the BW dataset in more depth is beyond the manuscript's scope.

[Figure]

Figure 1: Ozone absorption cross sections based on the SG16 and BW dataset for a selected temperature of -45°C (blue and violet curves). The red lines shows the relative difference between the SG16 and BW datasets for temperatures of -45 °C and -70 °C. The Dobson (light grey) and Brewer (dark grey) slit functions are shown as well.

[Figure]

Figure 2: (Same as Fig. 8 of the manuscript) Monthly mean difference between Dobson 104 and Brewers 010/226 (solid and dashed lines), at Hohenpeissenberg and using the Bernhard slit approximation. The different colors represent the results for the different ozone absorption cross section datasets.

Figure 5, 7 & Table 6: Using Amundsen-Scott as an example, it is obvious that daily Teff from TEMIS could have larger differences than its climatology results. I.e., for Oct. 2019, the difference between daily Teff (blue line) and climatology (dashed blue line) could be as much as 20 degrees. For Alert, the difference could also be as large as 10 degrees. In Fig 7, the Rel RIC difference for the Amundsen-Scott example is more than -2%, beyond the 2 sigma level. It looks like a similar issue for Alert in 2019 January.

Also, if we are using 1990-2020 climate to correct observations from Dobson records from 1960-70, will that introduce some more errors? The value of long-term Dobson records is for ozone recovery trend studies. Any suggestions?

We feel that the existing Figures and text appropriately show the limitations of using climatological temperatures. The reviewer has a point, because long-term temperature changes / trends could pose a problem. This problem, e.g. possible spurious ozone column trends, were pointed out also by reviewer #3. As indicated in our response to reviewer #3, we have added a paragraph in Sect. 3.4 (Uncertainties) to explain and quantify such potential problems.

Line 555-556: For Dobson data on WOUDC, the daily ozone values can be traced back to 1926, which is sparse anyway. However, observations in the 1960s and 70s already jumped to 15000 total observations each year, which is similar to the 1980s-2000s levels. Do you think using Teff climatology that only covers 1990-2020 will have any impact on corrections from the 1920s to the 1980s? Such impact on long-term ozone trend analysis should be at least mentioned and maybe discussed.

See response to the previous comment.

Line 560: To be practical, I fully agree with the authors such work should be carried out by WOUDC. Also, do you think there should be two versions of Dobson data published on WOUDC for the transition period? Anyway, the WOUDC team should be included in future discussions with the Dobson ozone cross-section committee.

In general, for future references and other eventualities, we think that the historical data as it currently stands should still reside at WOUDC. The new, reprocessed data should be easily distinguishable from the historical data (e.g. using different version numbers). During a transition period, we think it might be very valuable to publish two versions side by side, allowing easy comparison. We fully agree with the reviewer that the WOUDC team needs to be actively involved. We think the technical discussions about implementation at the WOUDC should take place outside the manuscript.

**Author response to comments of referee #3**

https://doi.org/10.5194/amt-2023-220-RC2

**Reviewer #3 general comments**

(1) In figures 5 and 7 I think it would be very valuable to add at least one station from the southern hemisphere mid-latitudes to show the global applicability of your results. If there is space, it would also be helpful to show a site closer to the equator than Izana with an annual cycle of a different shape.

We included the Southern Hemisphere station Lauder, NZ, in the Figure. It has a similar latitude to Hohenpeissenberg. We also included a station close to the Equator (Kampala). To keep the Figures halfway clear, we moved the findings for Izana to the supplementary material.

(2) You don't discuss trends. An objection to using the climatology is that, in theory, a long-term systematic trend in either stratospheric temperature, or the relative vertical distribution of ozone, would introduce a spurious trend in total ozone values. Presumably you have looked at this and found that the effect would not be significant, however I think it would be worth adding a sentence or two on this subject.

Good point. We have added a few sentences to the discussion of uncertainties in Section 3.4.

Trends of ozone effective stratospheric temperature are caused by trends in temperature and ozone profile. For the 1980 to 2000 period, effective temperature trends are of the order of −0.5 to −1 K per decade. For the 2000 to 2020 period, temperature trends are generally smaller. Assuming a fixed temperature (standard processing), or a climatological temperature (our suggested improved processing), does result in spurious trends of total column ozone. For a Dobson, these spurious total ozone trends are smaller than 0.1% per decade. Compared to the overall uncertainty (about 2%) this is small to negligible. For a Brewer, the spurious total ozone trends are almost an order of magnitude smaller, and are completely negligible.

(3) There is a comment I found odd in tables 3 and 4 in which it seemed the various co-authors obtained different answers for the same calculation, which should be clarified.

The various co-authors calculated the values independently, using different rounding methods, different software packages (Matlab, Python), and possibly slightly different values for input parameters (like Boltzmann's or Avogadro's constant, atmospheric pressure, humidity, …). This results in very minor differences in the obtained values, typically less than 0.03%.

Our intention was to show that slightly different values might be obtained when others repeat our calculations. However, to avoid possible confusion (as pointed out by the referee), we now simply drop the remark and have deleted the confusing Gröbner et al. numbers from Tables 3 to 5.

(4) The use of multiple wavelengths to largely eliminate the effect of aerosols is presented only very briefly (lines 96 and 97) and would not be able to followed by someone new to the field. I would prefer you add just a little bit more detail and ideally, show why the aerosol terms do not algebraically interfere with the later formulations, for example, equation 12 on line 315.

Good point. We have added sentences after line 99 to explain this a bit better:

Note that these weights fulfill the equations $\sum w_i \lambda_i^0 = 0$ and $\sum w_i \lambda_i^{-1} \approx 0$. This means that extinction by processes with a wavelength dependence of $\lambda^0$ or $\lambda^{-1}$, i.e., typical cloud or aerosol scattering, cancels out and would not affect the retrieved total ozone columns. See also Redondas 2018 (note the missing exponent -1 in their Eq. 6)

**Specific comments**

Lines 184-194 (Section 2.3.2) How well does the dispersion test really work for Brewers? The comment about Redondas et al. is a little bit sketchy compared to the Dobsons. I know you touch on this point again later. The broader question here is whether the general applicability of the results to the global network is compromised by large variations in the slit functions of operational instruments? Again I realize that you return to this point later on.

Generally, the dispersion test seems to work pretty well, and according to Redondas et al. (2018) there is no general need for the characterization of individual Brewer instruments with a tunable source. They demonstrated that errors in the range of 0.8% occur with different slit approximations or measured slit functions. According to Redondas et al. (2014), applying a standard absorption coefficient to a set of 33 different Brewer instruments, the error in Total

Ozone Column (TOC) would be less than 0.5%. These findings are consistent with the uncertainty estimation in Section 3.4 of our manuscript. In this section, we also show that the improvement resulting from the new cross-sections, including effective temperature, is significantly larger than potential errors introduced by the used slit functions (whether Brewer or Dobson).

We feel the discussion of the slit functions, and their impact on the accuracy of the calculated ozone absorption cross-section, is covered sufficiently in Sections 2.6 and 3.4, later in the manuscript. Hence, we would like to maintain Section 2.3.2 in its current form.

Lines 249-251 This formula takes into account the temperature, pressure, humidity and CO2 concentration, I believe – how do you determine these inputs?

We used standard values (e.g. 400 ppm for CO2 concentration, 1013.25 hPa for pressure, 0.5 for relative humidity, and 15°C for temperature) for the conversion. Sensitivity checks using potential value ranges (such as a temperature variation of 15°C and pressure levels between 900 and 1000 hPa) indicated that variations in the input parameters have very little impact on the results (<0.1%). Using more specific values for these input parameters is not necessary.

We added a couple of sentences in the manuscript to clarify this.

Line 290 – Figure 4 caption - "OS-LIDAR minus TEMIS" please clarify whether this is "daily" or "climate" for both.

Daily values are displayed for OS-LIDAR minus TEMIS. We updated the figure to clarify this.

Line 296 – Would it be possible to also include a southern hemisphere site at a similar latitude to MOHp? (This shouldn't clutter up the plot too much because it will be out of phase) The day-to-day variability would be lower in the SH than the NH I expect. If you have space it would be good to also include an equatorial site. Izana is at a relatively low latitude but clearly still has a NH mid latitude type seasonal cycle, albeit of smaller amplitude.

Done. See also answer to general comment #1.

Line 331 – I don't understand what you mean by "independent evaluation by co-author" – are you saying the different authors calculated the same quantity but came up with different numbers? That would be strange, and something I have never seen in a published paper before! I suggest trying to resolve the discrepancy and only including one value in the table. If the method was slightly different than that should be explained to the reader. Or, do you mean these are the values previously given in Gröbner et al. (2021)? Then it would not be surprising if there were slight differences.

Done. See also answer to general comment #3.